TOPICAL REVIEW

# Extracellular vesicles and their microRNA cargo in ischaemic stroke

Josie L. Fullerton, Caitlin C. Cosgrove, Rebecca A. Rooney and Lorraine M. Work 

*Institute of Cardiovascular and Medical Sciences College of Medical, Veterinary and Life Sciences, University of Glasgow, Glasgow, UK*

Handling Editors: Ian Forsythe & Harold Schultz

The peer review history is available in the Supporting Information section of this article (https://doi.org/10.1113/JP282050#support-information-section).

**Josie Fullerton** obtained her BSc in Neuroscience and MRes in Biomedical Science at the University of Glasgow. She then completed her PhD at the University of Strathclyde. **Caitlin Cosgrove** graduated from University of Glasgow with a BSc (Hons) in Human Biology. She is now in the MRes rotation year of her British Heart Foundation (BHF)-funded PhD programme during which she hopes to pursue further research relating to extracellular vesicle (EV)-derived microRNA in ischaemic stroke. **Rebecca Rooney** is a final year doctoral candidate in Cardiovascular Science at the University of Glasgow funded by the BHF investigating the role of EVs after ischaemic stroke. **Lorraine Work** has a research team based at the University of Glasgow, UK in the Institute of Cardiovascular and Medical Sciences. They are determining the potential of harnessing EVs loaded with therapeutic cargo in the setting of ischaemic stroke. She obtained her PhD in cardiovascular pharmacology from the University of Strathclyde and has been a PI for 15 years.

The Journal of Physiology

**Abstract** Acute ischaemic stroke (AIS) is a leading cause of death and disability. MicroRNAs (miRNAs) are short non-coding RNAs which hold the potential to act as a novel biomarker in AIS. The majority of circulating miRNAs are actively encapsulated by extracellular vesicles (EVs) produced by many cells and organs endogenously. EVs released by mesenchymal stem cells (MSCs) have been extensively studied for their therapeutic potential. In health and disease, EVs are vital for intercellular communication, as the cargo within EVs can be exchanged between neighbouring cells or transported to distant sites. It is clear here from both current preclinical and clinical studies that AIS is associated with specific EV-derived miRNAs, including those transported via MSC-derived EVs. In addition, current studies provide evidence to show that modulating levels of specific EV-derived miRNAs in AIS provides a novel therapeutic potential of miRNAs in the treatment of stroke. Commonalities exist in altered miRNAs across pre-clinical and clinical studies. Of those EV-packaged miRNAs, miRNA-124 was described both as an EV-packaged biomarker and as a potential EV-loaded therapeutic in experimental models. Alterations of miRNA-17 family and miRNA-17-92 cluster were identified in preclinical, clinical and MSC-EV-mediated neuroprotection in experimental stroke. Finally, miRNA-30d and -30a were found to mediate therapeutic effect when overexpressed from MSC and implicated as a biomarker clinically. Combined, EV-derived miRNAs will further our understanding of the neuropathological processes triggered by AIS. In addition, this work will help determine the true clinical value of circulating EV-packaged miRNAs as biomarkers of AIS or as novel therapeutics in this setting.

(Received 21 December 2021; accepted after revision 15 March 2022; first published online 14 April 2022)

**Corresponding author** L. M. Work: Institute of Cardiovascular and Medical Sciences, University of Glasgow, BHF GCRC, 126 University Place, Glasgow G12 8TA, UK. Email: Lorraine.Work@glasgow.ac.uk

**Abstract figure legend** Acute ischaemic stroke (AIS) is a leading cause of death and disability. Current preclinical and clinical studies indicate that AIS is associated with specific extracellular vesicle (EV)-derived miRNAs, including those transported via mesenchymal stem cells (MSC; MSC-derived EVs). Across current AIS preclinical and clinical literature, parallels of altered EV-packaged miRNAs exist; miRNA-124 was described as an EV-packaged biomarker and a potential therapeutic cargo in EVs in experimental stroke models. Alterations of miRNA-17-family and miRNA-17-92 cluster were identified in preclinical, clinical and MSC-EV-mediated neuroprotection in experimental stroke. Finally, miRNA-30d and -30a were found to mediate therapeutic effect when overexpressed from MSC and implicated as a biomarker clinically. Combined, EV-derived miRNAs will advance our understanding of the neuropathological mechanisms triggered by AIS. Additionally, this work will help ascertain the true clinical value of circulating EV-packaged miRNAs as biomarkers of AIS or as novel therapeutics in this setting.

## Introduction

**Acute ischaemic stroke.** Across the globe, stroke is the third-leading cause of death and disability combined (Feigin et al., 2021). In 2019, stroke was responsible for ∼6.55 million deaths worldwide with ischaemic stroke accounting for 62.4% of all stroke incidents (Feigin et al., 2021). Acute ischaemic stroke (AIS) occurs due to large artery atherothrombosis or cardiac embolism, which prevents blood flow to the brain and leads to rapid neurological deterioration (Virani et al., 2020).

At present, AIS diagnosis primarily relies upon clinical assessment, supplemented by neuroimaging. However, within the first hours of suspected stroke, clinical assessment can prove challenging, and if delayed, slowed diagnosis can lead to adverse clinical prognosis, increased risk of AIS reoccurrence and raised mortality rates (Liberman et al., 2020). Currently, there are no clinically available predictors or biomarkers of AIS and these could greatly improve patient prognosis by accelerated diagnosis and therapeutic intervention. Therefore, it is critical to develop a simple, non-invasive test to rapidly confirm the presence of suspected AIS and to improve current therapeutic interventions. MicroRNAs offer the promise to address both these needs through their potential both as a biomarker and as a novel therapeutic.

**MicroRNAs.** MicroRNAs (miRNAs) are small, non-coding, single-stranded RNAs that play a vital role in health and disease (Bhalala et al., 2013). Two mature miRNA species can be generated from the $5'$ and $3'$ arms of pre-miRNA, termed miRNA-5p/3p. Predominantly, one species remains while the other is degraded, yet co-existence of both species is increasingly reported (Griffiths-Jones et al., 2011; Ro et al., 2007).

At the post-transcriptional level, miRNAs regulate gene expression by modulating target messenger RNA resulting in altered levels of target protein(s). A single miRNA holds the potential to regulate thousands of downstream target genes, influencing entire gene networks and protein synthesis (Lai & Macleod 2005). These miRNAs are presents in all types of body fluid, including serum, plasma, urine and cerebrospinal fluid (CSF) (Weber et al., 2010). They are highly stable, relatively tissue-specific, associated with pathological states and readily detected, manipulated and measured; therefore, they are considered a promising and sensitive biomarker for a wide spectrum of diseases (Kosaka et al., 2010). Importantly, evidence indicates that circulating miRNAs are actively packaged and transported by extracellular vesicles (EVs) (Creemers et al., 2012; Gallo et al., 2012).

**Extracellular vesicles.** Based on their biogenesis, EVs can be broadly divided into exosomes (30–100 nm), microvesicles (50–1000 nm) and apoptotic bodies (50–2000 nm). These exocytotic particles are present in all biological fluids and are composed of a phospholipid bilayer and encapsulated cargo, which can include nucleic acids (including miRNAs), membrane-bound proteins, lipids, receptors, cytosolic metabolites, cytosolic and anti-apoptotic proteins as well as mitochondria. In both health and disease, EVs are essential for cross-cellular communication as they actively secrete and mediate cargo exchange between cells (Colombo et al., 2014).

Ever-expanding studies indicate that EVs target recipient cells such as natural killer cells, B and T cells, fibroblasts, endothelial cells, monocytes, macrophages, lymphocytes, adipocytes, hepatocytes, dendritic cells, cancer cells and glioblastomas, neural cells, microglia, astrocytes and oligodendrocytes (reviewed in Berumen Sánchez et al., 2021; Paolicelli et al., 2019). The majority of evidence suggest that EVs are internalised into these recipient cells by endocytosis; however, the exact mechanism of EV uptake remains unclear. Suggested mechanisms include clathrin-mediated endocytosis, caveolin-dependent endocytosis, lipid raft-mediated endocytosis, macro- or micropinocytosis, and phagocytosis; this uptake leads to subsequent release of EV contents from originating cells into recipient cells (reviewed in Margolis & Sadovsky 2019; Prada & Meldolesi 2016).

Due to their unique biology and essential role in cellular communication, EVs have attracted significant interest that has been further amplified by their therapeutic potential. As EVs derive their cargo from the contents of their originating cell, they are an attractive source of biomarkers for a variety of diseases. Recently, there has been a marked increase in knowledge of the role of EV-derived miRNAs in pathological processes, especially during cancer initiation and progression (Sun et al., 2020). Furthermore, emerging preclinical and clinical evidence

suggests an association between circulating levels of EV-encapsulated miRNAs in the setting of AIS.

**Mesenchymal stem cells.** Mesenchymal stem cells (MSCs) are the most prolific producers of EVs (Yeo et al., 2013) and it is through secretion of EVs and their bioactive cargo that MSCs are thought to exert much of their neurorestorative effects following AIS (Davis et al., 2021). miRNAs are among the functional cargo within MSC-EVs and have been of particular interest (Xin et al., 2014). Their ability to alter gene expression in target cells and influence many fundamental cellular processes including development, differentiation, homeostasis, communication and apoptosis (Vidigal & Ventura 2015) make them likely to play a key role in neurorestoration. This has culminated in preclinical studies in which miRNAs associated with post-stroke reparative mechanisms have had their expression altered in MSC-EVs. Furthermore, modified MSC-EVs have been administered to AIS animal models and their effect on recovery observed.

In this review, the primary aim is to establish current preclinical and clinical evidence that supports the association of AIS with individual EV-derived miRNAs, including those transported via MSC-derived EVs. Secondarily, we review current evidence from studies that modulated the levels of specific EV-derived miRNAs in AIS in order to investigate the novel therapeutic potential of miRNAs in the treatment of stroke. Combined, this review highlights specific EV-derived miRNAs that may play a key role in understanding the neuropathological processes triggered by AIS. In turn, this will help determine the true clinical value of circulating EV-packaged miRNAs as biomarkers of AIS or as novel therapeutics in this setting.

## Preclinical studies of EV-derived miRNAs on ischaemic stroke

Over the past decade, the therapeutic potential of EV-derived miRNAs in preclinical ischaemic stroke models has been assessed. The benefit of utilising preclinical models of stroke to assess the role of EV-derived miRNA is twofold: the biomarker profile of miRNAs actively loaded into EVs can be determined post-stroke, and secondly, specific miRNAs can be directly targeted using EVs as a therapeutic vehicle to up- or down-regulate miRNA expression, and following this neurological function and stroke-associated pathologies are determined (Fig. 1).

**Biomarker profiling studies of altered miRNAs in EVs after experimental stroke.** Several studies have determined circulating EV-packaged miRNA profiles

after experimental stroke. To determine the effect of middle cerebral artery occlusion (MCAO) duration on EV-packaged miRNA expression, male Sprague–Dawley (SD) rats were subjected to transient MCAO (tMCAO) for 5 min, 10 min or 2 h (Li et al., 2018). EVs were isolated from CSF and plasma, and miRNA profiling was performed on plasma-derived EVs before selected miRNAs were validated in both plasma and CSF. After 2 h tMCAO, there were 38 up- and 25 downregulated miRNAs. Expression profiles were different for 2 h tMCAO compared to 5 and 10 min tMCAO; miRNA-122 and miRNA-300 were found to be similarly expressed in both CSF and plasma EVs but were differentially expressed between sham and 5 or 10 min tMCAO. These preclinical results suggest that miRNA-122 and miRNA-300 could be used as diagnostic markers of ischaemic stroke, including transient ischaemic attacks (Li et al., 2018).

The findings of an unbiased screen of miRNAs in EVs isolated from serum of stroke patients were back-translated to preclinical models of stroke and cerebral small vessel disease (SVD) (van Kralingen et al., 2019). Three miRNA-17 family members (miRNA-17, -20b and -93) were upregulated in EVs from stroke patients with changes being driven by SVD rather than AIS. In preclinical models, there was an increase in expression of miRNA-17 and -93 between control, Wistar–Kyoto rats and the spontaneously hypertensive stroke prone rats (SHRSP), a widely accepted spontaneous model of SVD (Bailey et al., 2011; Bailey et al., 2014). Following experimental stroke in the SHRSP, no further increases in circulating EV-packaged miRNA-17 family

miRNAs were evident. Combined with results from clinical data, this suggests the alteration in EV-derived miRNA-17 family expression may be related to SVD rather than AIS (van Kralingen et al., 2019).

Using a combination of *in vivo* and *in vitro* research and bioinformatics, possible neuroprotective implications of EV miRNA expression and exercise after ischaemic stroke were described (Huang et al., 2021). Male SD rats were separated into four groups: physical exercise (PE) with or without AIS and sedentary (Sed) with or without AIS. Rats were subjected to tMCAO and exercise training for 28 days, after which plasma and CSF were harvested and miRNAs profiled. A total of 16 miRNAs were upregulated and 25 downregulated in PE and PE-AIS compared to Sed and Sed-AIS. Of these miRNA-92b and -370 were upregulated and miRNA-136, -665 and -3068 downregulated, demonstrating a significant fold change in both PE *vs*. Sed and PE-AIS *vs*. Sed-AIS groups. The Kyoto Encyclopaedia of Genes and Genomes database was used to understand the biological role of these up- and downregulated miRNAs, with many of the signalling pathways being associated with neurological impairment or recovery post stroke. An interaction network identified their possible role in exercise-induced neuroprotection, which ranged from learning to vasodilatation. All of these up- and downregulated miRNAs play a role in stroke pathology and thus impact recovery (Huang et al., 2021).

**Therapeutic modulation of miRNAs using EVs after experimental stroke.** The numerous roles of EV-derived miRNAs are still being determined in both preclinical

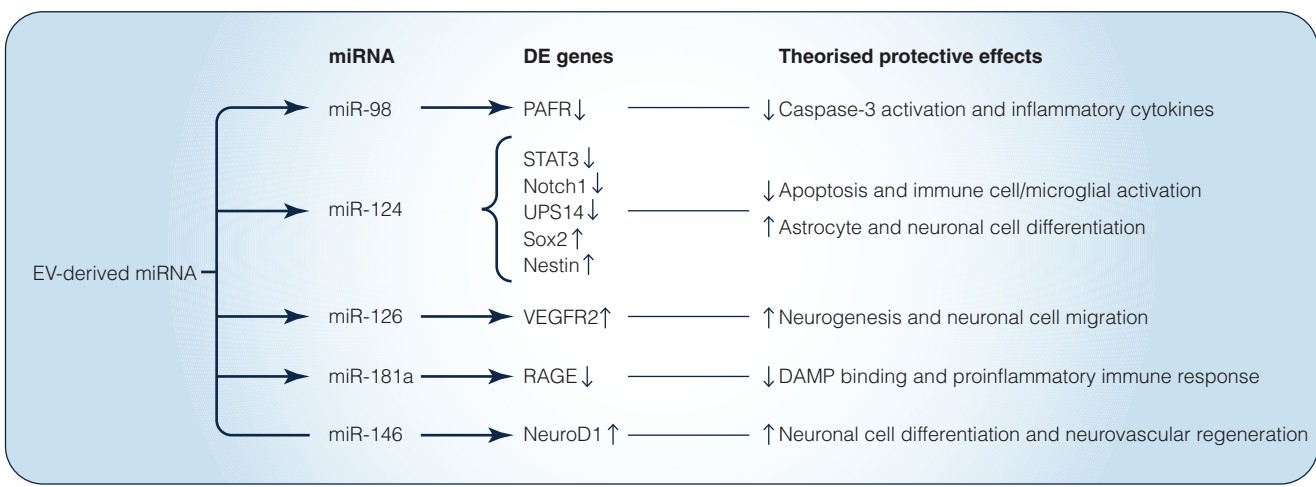

**Figure 1. Protective effects of altered EV miRNA cargo in preclinical ischaemic models of stroke**
Flow diagram detailing altered miRNA cargo in *in vitro* and *in vivo* preclinical stroke simulation. Current literature shows that modulating levels of miRNA-98, -124, -126, -146 and -181a leads to either upregulation or down-regulation of certain target genes and their subsequent theorised protective roles in ischaemic stroke. DAMP, damage-associated molecular pattern; DE, differentially expressed.

and clinical studies. However, understanding the effects of modifying miRNA expression within EVs for therapeutic purpose is a relatively new aspect within this field. In theory, by altering miRNA cargo, miRNA-enhanced EV populations could be used to improve stroke outcome (Fig. 1). By identifying miRNAs that positively affect stroke outcome preclinically, such as those listed here, novel EV-derived miRNA therapies could be used clinically.

*miRNA-124.* miRNA-124 is highly expressed in the brain (Sun et al., 2015) and has previously been reported to be a therapeutic target after ischaemic stroke preclinically (Hamzei Taj, Kho, Aswendt et al., 2016; Hamzei Taj, Kho, Riou et al., 2016). The severity of ischaemic stroke has been linked to circulating plasma miRNA-124 levels (Ji et al., 2016; Qi et al., 2021). miRNA-124 in EVs has also been targeted therapeutically following identification of its evident protective role in interleukin-4 (IL-4)-induced M2 BV2 microglial cells (Song et al., 2019). Intravenous delivery of M2-derived EVs following tMCAO in mice led to a decrease in infarct volume, fewer apoptotic cells and improved functional outcomes compared to phosphate-buffered saline (PBS) control determined 3 days after tMCAO (Song et al., 2019). Bioinformatic analyses using miRBase identified several altered target genes including the deubiquitinating enzyme *UPS14*, which was downregulated after M2-EV treatment. To further understand the relationship of miRNA-124 and UPS14, mice were treated with M2-EVs, miRNA-124-kd (knockdown)-EVs, miRNA-124-kd-EVs plus USP14 inhibitor, or miRNA control EVs before stroke. While M2-EVs significantly reduced infarct volume compared to PBS control, miRNA-124-kd EVs increased infarct volume with poorer neurological outcome determined using a modified neurological severity score (mNSS). The USP14 inhibitor reversed the deleterious effects of the miRNA-124-kd EVs and decreased infarct volume and improved mNSS. Combined, these results indicate that M2-EV-derived miRNA-124 may play a protective role after ischaemic insult by downregulating *UPS14* (Song et al., 2019).

A follow-on study (Li, Song et al., 2021) added further mechanistic insight into the protective effect of miRNA-124 in M2-EVs after experimental stroke. Male ICR mice were divided into four groups, sham, PBS control, M2-sEV and miRNA-124-kd, subjected to tMCAO and EVs were delivered intravenously with neurological assessments on days 7 and 14. As previously described (Song et al., 2019), this study confirmed that ischaemic lesion volume was reduced and neurological outcome, determined using a mNSS and the corner test, improved after tMCAO in M2-sEVs treatment group compared to miRNA-124-kd EV treatment (Li, Song et al., 2021). Mechanistically, glial fibrillary acidic protein

(GFAP) staining revealed that M2-sEVs reduced scar area/thickness. Conversely, miRNA-124-kd EVs increased glial scar formation, suggesting that miRNA-124 plays a role in scar formation after stroke. Interestingly, mice treated with M2-sEVs showed an increase in Sox2 transcription factor (days 7 and 14) while nestin levels were increased only at day 7 post-tMCAO (Li, Song et al., 2021). Sox2 transcription factor allows for the transformation of neuroblasts from astrocytes, while nestin is a marker of neural progenitors (Niu et al., 2015). These findings suggest astrocytes may be transformed into neural progenitors after ischaemic insult. By utilising miRBase, two target proteins of miRNA-124 were identified as signal transducer and activator of transcription 3 (STAT3) and pSTAT3. Western blot revealed that expression of both proteins was significantly increased in the infarct region of tMCAO brains 7 days post-stroke and M2-sEV treatment reduced their expression while miRNA-124-kd EVs upregulated STAT3 and pSTAT3 expression compared to M2-sEVs. Together, this suggests that miRNA-124 within M2-sEVs blocks STAT3 activation. This, coupled with reduced Notch1 expression after tMCAO in M2-sEV-treated mice, further support the theory that M2-sEVs prevent astrocyte differentiation whilst promoting transformation of astrocytes to neurons (Li, Song et al., 2021). Hence, miRNA-124 appears to have benefit in the setting of experimental stroke through targeting several key pathways underlying lesion progression supporting the theory that miRNAs offer the potential to achieve a polytherapy style outcome from a single intervention.

*miRNA-126.* The role of EV-packaged miRNA-126 has been demonstrated *in vivo* and *in vitro* (Chen et al., 2015; Wang et al., 2020). EVs derived from cultured human endothelial cells subjected to oxygen–glucose deprivation (OGD) were shown to have significantly reduced miRNA-126 expression compared to normoxic control EVs (Chen et al., 2015). Additionally, in samples from male Wistar rat serum, EV-derived miRNA-126 was reduced at 3 h post-tMCAO and permanent MCAO (pMCAO) returning to levels similar to that of baseline by 24 h (Chen et al., 2015). Looking at diabetes as a stroke-associated comorbidity, *db/db* type II diabetic mice were subjected to pMCAO and treated intravenously with either PBS or bone marrow-derived endothelial progenitor cells (EPC)-EVs transfected with miRNA-126 mimic or scrambled control (Wang et al., 2020). DiL-labelled EVs showed localisation to endothelial cells, neurons, astrocytes and microglia within the brain after pMCAO. Infarct volume (day 2 or 14) was reduced in EPC-EV-miRNA-126-treated mice compared to those given PBS or EPC-EV scrambled. Cerebral blood flow and microvascular density was better preserved in EPC-EV-miRNA-126-treated mice compared to

EPC-EV scrambled with an accompanying improvement in neurological function. Levels of angiogenesis and neurogenesis in the peri-infarct region were higher with EPC-EV-miRNA-126 compared to EPC-EV scrambled. Vascular endothelial growth factor receptor 2 (VEGFR2) expression was significantly increased with EPC-EV-miRNA-126 treatment compared to both EPC-EV and control. Combined, these results suggest that EV delivery of miRNA-126 may lead to increased angio- and neurogenesis after ischaemic insult by upregulating VEGFR2 expression (Wang et al., 2020).

*miRNA-98.* In the brain, the immune system is thought to play a significant role in post-stroke outcome (Fu et al., 2015). The role of miRNA-98 in microglial phagocytosis in ischaemic stroke models was demonstrated in C57BL/6J mice using adeno-associated virus (AAV) vectors to label both neuronal EVs and miRNA-98 prior to experimental stroke (Yang et al., 2021). Co-localisation identified the presence of miRNA-98 and the EV marker CD63 within neurons. Three weeks after virus injection mice were subjected to tMCAO. After 6 h reperfusion CD63[+] miRNA-98 containing EVs were found to be inter-nalised by microglia within the ipsilateral hemisphere demonstrating that neuronal derived EVs containing miRNA-98 can be transferred to microglia. Platelet activating factor receptor (PAFR) was confirmed as a target of miRNA-98 (Yang et al., 2021). PAFR has been previously shown to increase microglial phagocytosis in other neurological diseases (Bellizzi et al., 2016) and so may serve as an intercellular mediator involved in the communication between neurons and microglia after ischaemic stroke (Yang et al., 2021).

*miRNA-146.* To determine the contribution of miRNAs to the response to electro-acupuncture (EA) after ischaemic stroke, male SD rats were divided into three surgical groups: sham, tMCAO, and tMCAO+EA (Zhang et al., 2020). Markers of EVs were increased in tMCAO+EA compared to tMCAO alone and were increased in tMCAO alone compared to sham. miRNA profiling of peri-ischaemic striatum-derived EVs of tMCAO and tMCAO+EA rats identified 25 differentially expressed miRNAs. Validation using qRT-PCR confirmed miRNA-30a, -125b and -146b were upregulated, while miRNA-1949 was down-regulated. Focusing on miRNA-146b, implicated in neural stem cell differentiation (Xiao et al., 2015), further studies determined the effect of inhibiting miRNA-146b after tMCAO±EA. Immunofluorescence staining identified the number of neonatal neurons within the peri-ischaemic striatum subventricular zone (SVZ) decreased with miRNA-146b inhibitor treatment compared to tMCAO, whereas the number increased with tMCAO+EA treatment compared to tMCAO. The miRNA-146 inhibitor reversed the beneficial effect of EA after tMCAO. In addition, expression of NeuroD1, a transcription factor known to potentiate reprogramming of other cell types into neurons (Pang et al., 2011), was increased in tMCAO+EA treatment compared to tMCAO and tMCAO+miR146b inhibitor treatment significantly decreased NeuroD1 compared to tMCAO+EA. Together, these findings suggest EA along with EV-derived miRNA-146b improves neuronal stem cell differentiation into neurons post-stroke (Zhang et al., 2020).

*miRNA-181a.* miRNA-181a has been found to be altered after ischaemic stroke and has been modulated therapeutically preclinically to positively affect outcome (Ouyang et al., 2012; Xu et al., 2015). Among the predicted targets for miRNA-181a is the receptor for advanced glycation end-products (RAGE). Hence, Kim et al. (2021) determined the role of anti-RAGE EVs loaded with antisense-181a oligonucleotides in ischaemic stroke. Male SD rats were subjected to tMCAO and those treated with EVs from HEK293T cells trans-fected with antisense miRNA-181a had significantly reduced infarct volume. miRNA-181a levels were reduced along with an increase in anti-apoptotic Bcl-2 and a concomitant reduction in levels of RAGE, apoptosis and the pro-inflammatory cytokine tumour necrosis factor $\alpha$. Immunostaining identified downregulation of RAGE with intranasal delivery of RBP-antisense miRNA-181a EVs. Combined, these results suggest that EVs loaded with anti-sense miRNA-181a reduced neuronal inflammation by inhibiting RAGE-induced inflammation after ischaemic insult (Kim et al., 2021).

## MSC-derived EVs in ischaemic stroke

MSCs are self-renewing multipotent stem cells (El-Hashash 2020) that have emerged over recent decades as a promising stroke therapeutic that can promote repair and regeneration of damaged cerebral tissue (Li, Shi et al., 2021). MSCs are the most commonly used stem cell in regenerative medicine due to their availability, lack of ethical controversy, low immunogenicity, and relative ease of isolation and culture (Musiał-Wysocka et al., 2019). Although recent systematic reviews and meta-analyses have highlighted the efficacy of MSC therapy for AIS in preclinical (Satani et al., 2019) and clinical trials (Li et al., 2020), improvements in neuro-vascular repair and functional recovery have been limited and results inconsistent, and therefore attention has turned to understanding MSCs' paracrine mechanism of action (Fig. 2).

EV-facilitated miRNA transfer was suggested as a mediator of MSCs' neuroprotective effects when significantly increased levels of miRNA-133b was shown in the brains of ischaemic rat models subjected to

intravenous MSC treatment (Xin et al., 2012). Further *in vitro* investigation identified increased levels of miRNA-133b in MSC-EVs exposed to ischaemic tissue extracts from rats. When neurons and astrocytes were cultured with these MSC-EVs they expressed high levels of miRNA-133b that was associated with increased neurite outgrowth suggesting that EVs mediated miRNA-133b transfer to neural cells (Xin et al., 2012). To test this theory *in vivo*, researchers altered miRNA-133b levels in MSCs and their corresponding EVs (Xin et al., 2013). Significant improvement in functional recovery, including adhesive removal and foot-fault test, was observed in the MSC-treated tMCAO group compared to control groups. These improvements were enhanced in the upregulated miRNA-133b-MSC-treated group but decreased in the downregulated miRNA-133b-MSC-treated group, suggesting miRNA-133b-enriched EVs mediated the therapeutic effects (Xin et al., 2013). In addition to functional improvement, MSC-EVs tagged with green fluorescent protein were detected in adjacent astrocytes and neurons in the ischaemic boundary zone confirming their interaction with target cells. Taken together these results provided experimental evidence that EV-mediated transfer of miRNA between MSCs and neural cells may be involved in promoting functional recovery and axonal plasticity post-stroke.

Other studies implicate miRNA-enriched EVs in mediating MSCs' therapeutic effects. MSC-EVs modified to overexpress miRNA-126 (miRNA-126+MSCs) promoted functional recovery when intravenously administered to rats after tMCAO (Geng et al., 2019). Compared to controls and downregulated miRNA-126 MSC-EVs, miRNA-126+MSCs significantly increased expression of von Willebrand factor and doublecortin, indicators of vasculogenesis and neurogenesis, respectively. Furthermore, miRNA-126+MSCs reduced microglial activation and regulated neuro-inflammation (Geng et al., 2019). Similarly, miRNA-210 is an important mediator of responses, such as apoptosis and metabolism in ischaemic conditions (Zhang et al., 2019). MSC-EVs overexpressing miRNA-210 were intravenously administered to tMCAO mouse models and enhanced angiogenesis via upregulation of VEGF signalling and improved animal survival rate (Zhang et al., 2019). EVs have also been modified to overexpress miRNA-124, the most abundant brain miRNA (Yang et al., 2017). When administered intravenously to ischaemic mouse models, miRNA-124 enriched EVs reduced ischaemic injury and

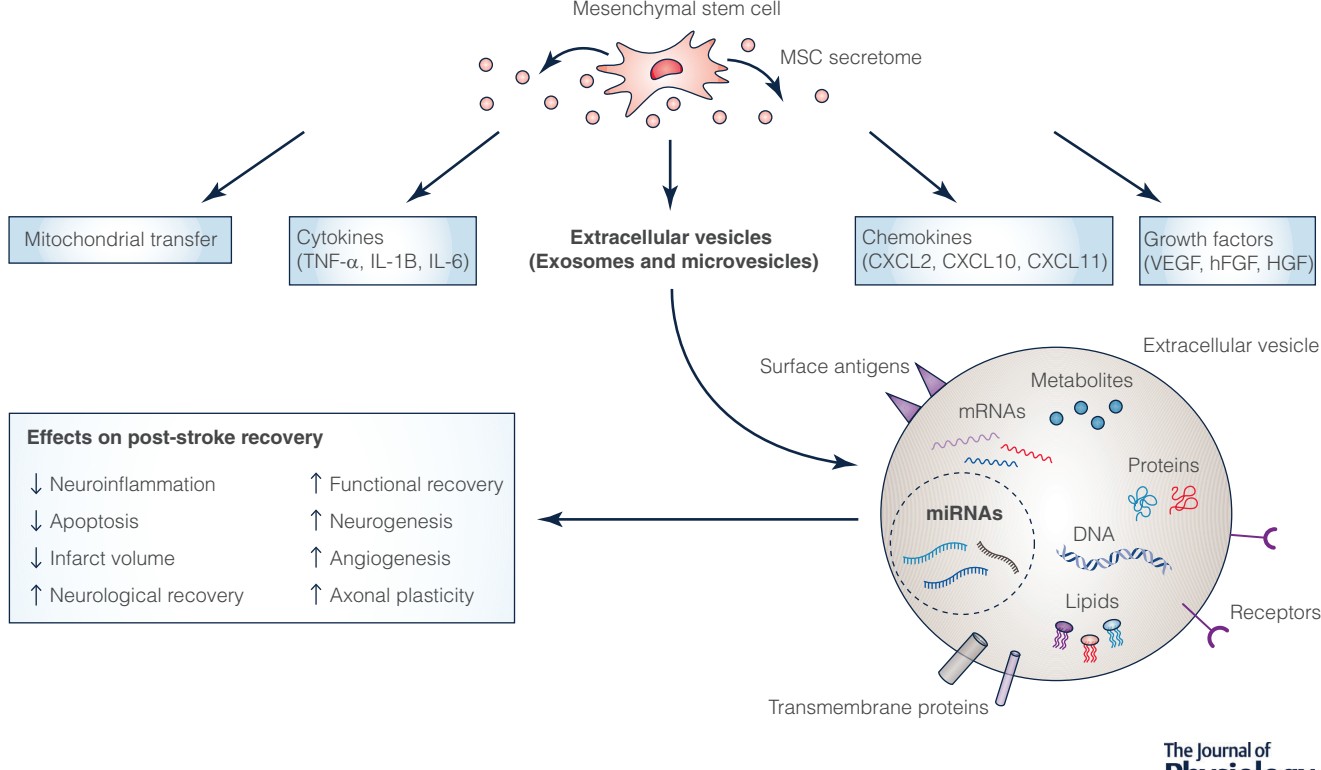

**Figure 2. Effects of mesenchymal stem cell (MSC)-derived EVs on post-stroke recovery**
Diagrammatic representation of MSC-EVs and their functional cargo highlighting the role of miRNAs in promoting post-stroke recovery.

enhanced neurogenesis by promoting the transition of neural progenitors to neuronal phenotypes at the infarct (Yang et al., 2017).

The EV-mediated transfer of miRNA-17-92 cluster has also been associated with enhanced functional recovery and neural plasticity post-stroke (Xin et al., 2017). Rats subjected to 2 h tMCAO received intravenous injection of miRNA-17-92 cluster-enriched EVs (miRNA-17-92+), control EVs or liposomes 24 h post-stroke. Behavioural tests, performed up to 28 days post-MCAO, found that EV treatments significantly improved neurological recovery compared to liposome treatment but that the effect was significantly greater in the miRNA-17-92+ EV group. Enhanced neurological recovery was associated with increased neurogenesis, oligodendrogenesis and neuronal plasticity (Xin et al., 2017). The mechanism underlying this recovery was discussed in a subsequent study (Xin et al., 2021) highlighting the role of miRNA-17-92 in downregulating expression of the phosphatase and tensin homologue (*PTEN*) gene and activating the phosphoinositide 3-kinase/Akt/mechanistic target of rapamycin pathway. Activation of this pathway enhances axon extension and myelination and improves the electrophysiological responses associated with improved functional recovery following miRNA-17-92+ EV administration (Xin et al., 2021).

Bioinformatic analysis revealed a gene regulatory pathway important in neurological recovery, the *LCN2* gene and its upstream regulator miRNA-138 (Deng et al., 2019). *LCN2* was found to be overexpressed in mouse models of experimental stroke. *LCN2* codes for an iron transport protein, neutrophil gelatinase-associated lipocalin, which is secreted by astrocytes under ischaemic conditions and promotes neuronal apoptosis and inflammation. When MSC-derived EVs overexpressing miRNA-138 were delivered to C57BL/6 mice after tMCAO, neuronal loss and infarct volume were reduced in the treated mice compared to control tMCAO mice after 4 weeks. Further, the presence of inflammatory factors and apoptotic marker proteins was significantly decreased in the brain (Deng et al., 2019). The study demonstrated that EV-mediated delivery of miRNA-138 downregulates *LCN2* expression and subsequently reduces neuroinflammation and neuronal apoptosis (Deng et al., 2019).

Several miRNAs identified in MSC-EVs affect outcome in experimental stroke through an effect on apoptotic pathways. miRNA-22 has been implicated in reducing neuronal apoptosis in both *in vitro* and *in vivo* models of ischaemic injury (Zhang, Liu et al., 2021). miRNA-22 was shown to be upregulated in MSC-EVs, and a miRNA-22 inhibitor removed MSC-EVs' protective effect against apoptosis. The mechanism suggested that miRNA-22 was transferred to neurons, where it bound and inhibited expression of *KDM6B*. This suppressed KDM6B binding

to *BMP2* and reduced *BMF* expression, which induced cell apoptosis (Zhang, Liu et al., 2021). MSC-EVs modified to overexpress miRNA-31 successfully reduced neuronal apoptosis and promoted neurological recovery after administration to mouse models 2 h after MCAO (Lv et al., 2021). miRNA-31 binds to *TRAF2*, which is responsible for promoting *IRF5* expression, which contributes to neuronal damage. MSC-EV-mediated transfer of miRNA-31 downregulates *TRAF2* and *IRF5* expression leading to reduced neuronal apoptosis (Lv et al., 2021).

The immunomodulatory properties of miRNAs in MSC-EVs have been utilised to reduce injury after experimental stroke. When neuronal cell damage is detected, microglia become activated in two forms: M2-like microglia have anti-inflammatory and restorative effects (Cherry et al., 2014); and M1-like microglia have pro-inflammatory effects which are important in the initial stages following brain injury, but when chronically activated they release inflammatory cytokines, chemokines and reactive oxygen species which exacerbate ischaemic damage (Patel et al., 2013). MSC-EVs overexpressing miRNA-30d reduced cerebral injury through inhibition of M1-like microglial polarisation while promoting M2-like polarisation (Jiang et al., 2018). By inhibiting polarisation of microglia to the M1-like phenotype, chronically induced secondary tissue damage can be reduced. Furthermore, promotion of the M2-like phenotype encourages neurorestorative effects and reduces inflammation. This was confirmed in a rat model of ischaemic stroke where intravenous treatment with EVs overexpressing miRNA-30d promoted M2-like polarisation and inhibited M1-like polarisation subsequently reducing cerebral injury (Jiang et al., 2018). Small RNA expression analysis and deep sequencing were performed on microglia exposed to OGD with or without MSC-EV treatment. miRNA-146a was one of the altered miRNAs with higher expression in the MSC-EV treatment group (Zhang, Zou et al., 2021). Mice subjected to ischaemic stroke were intravenously administered MSC-EVs or miRNA-146a knockdown MSC-EVs. Treatment with miRNA-146a knockdown EVs failed to significantly reduce infarct volume to the degree of the MSC-EV group thus implicating miRNA-146a in the attenuation of microglia-mediated neuroinflammation (Zhang, Zou et al., 2021).

To date, there is only one ongoing clinical trial assessing the safety and efficacy of allogenic miRNA-modified MSC-derived EVs in ischaemic stroke patients (https://clinicaltrials.gov/ct2/show/NCT03384433). This trial proposes to over-express miRNA-124 given its role in enhancing neurogenesis (Yang et al., 2017). EV therapy has the potential to overcome limitations presented by MSC-based approaches, such as vascular occlusion, tumour formation (Herberts et al., 2011) and entrapment

in the liver and lungs due to their size (Moon et al., 2019). Furthermore, the ease with which the miRNA content of MSC-EVs can be modified shows remarkable potential as a safe means of delivering these bioactive molecules to target cells to promote neurological repair mechanisms following ischaemic stroke (Alkaff et al., 2020). It is important to remember, however, that MSC-derived EVs can enhance post-stroke recovery without artificial miRNA modification (Satani et al., 2019; Li et al., 2020) suggesting other bioactive factors may also be at work. Additionally, the combinatory role of several miRNAs to promote recovery should not be undermined despite findings that MSC-EVs lose their protective abilities upon the knockdown of individual miRNAs (Deng et al., 2019; Geng et al., 2019; Xin et al., 2013; Zhang, Zou et al., 2021).

## Clinical studies of EV-derived miRNAs in ischaemic stroke

In the clinical setting, ever-increasing evidence suggests the involvement of EV-mediated cross-cellular communication via miRNAs following stroke. Significant alterations in the expression of specific EV-packaged miRNAs have been reported in patients with confirmed stroke when compared to controls; this literature supports the potential use of EV-derived miRNAs as a novel diagnostic and prognostic tool for ischaemic stroke (Fig. 3). Here, we will discuss those miRNAs found to be altered in circulating EVs after ischaemic stroke.

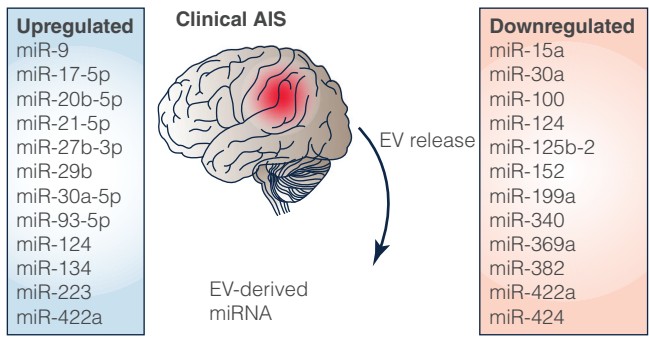

**Figure 3. Altered expression of EV-derived miRNAs across clinical AIS samples**
Diagrammatic representation of EV-derived miRNA expression extracted from AIS patient samples. Across studies, 12 EV-derived miRNAs were found to be upregulated and a further 12 downregulated following AIS. Three miRNAs of interest, namely miRNA-30a, -124, and -422a, were reported to be both upregulated and downregulated across the literature; this disparity may be due to factors such as specific 3′ targeting, sample collection time and patient characteristics.

The expression of brain-specific miRNAs is upregulated in serum-derived EVs following stroke, when compared to non-stroke controls. Within 24 h of symptom onset, the expression of miRNA-9, -124 and -134 was positively correlated to poor prognosis of stroke patients assessed through the National Institutes of Health Stroke Scale (NIHSS), infarct volume and serum levels of interleukin-6 (IL-6) (Ji et al., 2016; Zhou et al., 2018). Conversely, miRNA-124, also contained in serum-derived EVs, was reported to be significantly downregulated 2 h after stroke onset, and continued to decrease 6 h after onset (Qi et al., 2021). The altered expression of miRNA-124 exhibited high sensitivity in accuracy, diagnosis and prediction of early-stage ischaemic stroke, which was negatively correlated with NIHSS score and serum IL-6 levels. There are multiple factors that could influence the differential expression of miRNA-124 between studies, primarily the specific targeting of the 3′ strand of miRNA-124 by Qi et al. (2021), but also sample collection post-stroke (2 h *vs.* 24 h), patient characteristics or patient comorbidities.

Current evidence also highlights the expressional change of miRNAs linked to the inflammatory response triggered by ischaemic injury. Within 72 h of stroke onset, expression of miRNA-223 contained in serum-derived EVs was upregulated, compared to healthy controls (Chen et al., 2017). This alteration was positively correlated with NIHSS score, where stroke patients with poor outcomes tended to demonstrate higher miRNA-223 expression, compared to those with good outcomes. Further, this study reported that there were no changes in the expression of inflammatory-associated miRNA-21 and -145 from serum-derived EVs (Chen et al., 2017).

In ischaemic stroke, serum levels of EV-packaged miRNA-152 were significantly downregulated in patients with stroke when compared to healthy controls; this was most pronounced in those patients with NIHSS scores ≥7 (Song et al., 2020). In addition, miRNA-152 expression was most notably downregulated in samples from large-artery atherosclerosis (LAA) stroke patients, compared to small-vessel occlusion, cardioembolism (CE) and stroke of undetermined aetiology (SUA). Furthermore, serum EV levels of miRNA-152 were significantly lower in the acute phase (<24 h from symptom onset) compared to the chronic phase (>14 days) (Song et al., 2020). Combined, these data show that miRNA-152 expression is related to the severity of endothelial injury following ischaemic stroke.

Across a minority of clinical literature, analysis was further divided into symptom onset and sample collection time, namely: hyperacute ischaemic stroke (HIS, within 6 h), acute ischaemic stroke (AIS, days 1 to 7), sub-acute ischaemic stroke (SIS, days 8 to 14) and restorative ischaemic stroke (RIS, days >14). The expression of apoptosis-related miRNAs from plasma-derived EVs were found to alter across time points (Wang et al., 2018).

Specifically, in SIS and RIS the expression of miRNA-21 was upregulated when compared to controls, and miRNA-30a expression was upregulated in HIS and downregulated in AIS. A second study reported the downregulation of plasma-derived EV cargo miRNA-422a and -125b-2 in SIS patients (defined here as 4–14 days) and miRNA-422a expression was upregulated in AIS (days 1–3 from onset), when compared to controls (Li et al., 2017), while in SIS patients miRNA-422a and -125b-2 were dramatically downregulated, when compared to those with AIS. However, there were no marked differences in NIHSS scores between AIS and SIS patients. Together these data suggest that altered expression levels of miRNA-21, -30a, -125b-2 and -422a are promising biomarkers to distinguish between HIS, AIS, SIS and RIS; however, prognostic correlation was not assessed, and time points differed between studies (Li et al., 2017; Wang et al., 2018).

Ischaemic stroke can also be subdivided by specific stroke aetiology or lesion location. Previous work aimed to determine whether EV-contained miRNA levels were modified based on lesion topography in patients with cortical-subcortical stroke (CSC) or subcortical stroke (SC) (Otero-Ortega et al., 2021). EV-contained expression levels of miRNA-15a, -100, -199a, -369a and -424 were downregulated in CSC and SC patients, compared to healthy controls. The CSC group showed an even lower expression of miRNA-15a, -100, -339 and -424 compared with the SC group. At 3 months post-stroke, there was a correlation between downregulated miRNA-100 in circulating EVs and improved NIHSS score. Previous studies have linked miRNA-15a, -100 and -424 expression to improved vessel formation and Von Willebrand factor, which could enhance angiogenesis. Although infarct volume was measured, this was only assessed against the quantity of circulating EVs, rather than the specific miRNA cargo of EVs. In a sub-study, this group also demonstrated that circulating EV expression of miRNA-340 was downregulated and miRNA-29b was upregulated in stroke patients, when compared to healthy controls (Otero-Ortega et al., 2020).

As the majority of current literature compares patients with stroke to healthy non-stroke controls, it is not clear whether altered miRNA expression is reflective of stroke itself, associated risk factors or comorbid conditions. As previously mentioned, our group identified four miRNAs that were significantly altered in patients post-stroke compared to stroke mimics (non-stroke controls) (van Kralingen et al., 2019), in which participants were enrolled with suspected stroke and confirmed whether they had had a stroke or were a stroke mimic. Within 48 h of symptom onset, the expression of EV-packaged miRNAs from the miRNA-17 family (miRNA-17, -20b and -93) and miRNA-27b were upregulated. These alterations were primarily driven by an increase in patients with SVD stroke, as no significant differences in people with LAA, CE or SUA were seen. Therefore, the altered expression of miRNA-17 family may parallel the pathological processes behind cerebrovascular SVD rather than AIS itself (van Kralingen et al., 2019).

Across current literature, most studies have assessed EVs isolated from serum or plasma samples, using an approved EV isolation kit and characterised via transmission electron microscopy, nanoparticle tracking analysis and western blotting. However, differences in control group, sample collection time, type of sampling, EV isolation/storage and −3p/5p specification may account for disparity of results between studies (Chen et al., 2017; Wang et al., 2018). Moreover, stroke treatment may significantly alter EV cargo (Thulin et al., 2020). There are reported differences in the number of patients who received intravenous thrombolysis and mechanical thrombectomy between CSC and SC patients (Otero-Ortega et al., 2021).

At present, clinical literature has successfully identified and validated changes in circulating EV miRNA expression in people with stroke (Fig. 3). These data indicate that specific miRNA cargoes of serum- or plasma-derived EVs are associated with the severity of stroke, infarct volume and circulating level of IL-6. In addition, studies have further demonstrated pronounced expression alteration in specific subcategories of stroke, such as stroke aetiology, lesion location or time since symptom onset. These data demonstrate that microRNAs hold great potential as a diagnostic and prognostic tool to evaluate the degree of ischaemic injury in peripheral 'liquid' biopsies; however, further studies employing larger and more diverse populations with consistent methodologies are required to confirm the true clinical value of EV-contained miRNAs in ischaemic stroke.

## Commonalities in EV-packaged miRNA expression changes from bench to bedside

Previous systematic reviews of miRNAs and AIS have highlighted marked heterogeneity across preclinical (Fullerton et al., 2022) and clinical (Dewdney et al., 2018; Fullerton et al., 2022) studies in terms of the miRNAs linked to disease pathophysiology. Indeed, the failure to consider EV-packaged specific cargo was considered one contributing factor to the disparity in findings. Here, we describe the current literature for EV-packaged miRNAs in the setting of AIS (preclinical and clinical). Given the relatively new field, commonalities in altered miRNAs that have been identified across preclinical and clinical studies are already evident. Of those EV-packaged miRNAs, miRNA-124 was described both as an EV-packaged biomarker (Ji et al., 2016; Qi et al., 2021) and as a potential mediator of EV therapy in experimental models (Li, Song et al., 2021; Song et al., 2019; Yang et al., 2017).

Reduced miRNA-124 expression acutely (up to 6 h) after AIS negatively correlated to outcome (NIHSS) and IL-6 levels (Qi et al., 2021) while increased expression in circulating EVs at 24 h from symptom onset positively correlated with NIHSS and IL-6 levels (Ji et al., 2016). Likewise, therapeutically increasing miRNA-124 resulted in reduced infarct volume and improved neurological recovery while knockdown of miRNA-124 worsened outcome after experimental stroke (Li, Song et al., 2021; Song et al., 2019; Yang et al., 2017). Our own study demonstrated alterations in miRNA-17 family members (miRNA-17 and -93) across circulating EVs isolated from preclinical and clinical samples confirming a potential link with the chronic sequelae of SVD rather than AIS itself (van Kralingen et al., 2019). Further, members of the miRNA-17-92 cluster (which included miRNA-17) have been identified as key mediators in MSC-mediated neuroprotection in experimental stroke (Xin et al., 2017; Xin et al., 2021). Finally, two members of the miRNA-30 family (miRNA-30d and -30a) have been found to be therapeutic when overexpressed from MSC (Jiang et al., 2018) and implicated as biomarkers to distinguish between HIS and AIS clinically, respectively (Wang et al., 2018). Taken together, that so many common altered EV-packaged miRNA changes have already been described demonstrates the strength both in profiling EV-packaged miRNA cargo, and not simply circulating changes, and further in the unique potential of using EVs loaded with miRNA as a novel therapeutic for AIS.

## Conclusion

Evidence from preclinical and clinical studies for the involvement of EVs and their miRNA cargo in providing a potential biomarker for AIS is strong. Further, their use in preclinical studies as a novel therapeutic and as a mediator in the beneficial effects of stem cells is mounting. If successfully translated to humans, this could prompt the development of a novel neuroregenerative therapeutic strategy for stroke through EV therapy. The systemic administration of EVs, rather than stem cells for example, has the major advantage of being able to deliver the therapeutic agent across the blood–brain barrier (BBB) (Heidarzadeh et al., 2021). The exact mechanism of this ability is largely unknown but recent studies have highlighted a diverse range of uptake strategies across the intact BBB including the use of specific transporters, initiators of adsorptive transcytosis, receptor interactions, and a notable influence from the brain-to-blood efflux system. Additionally, inflammation, such as that in stroke, breaks down the BBB and therefore increases its permeability, which may further enhance these transport processes (Banks et al., 2020). Another advantage is that EVs would avoid the risks associated with cell-based therapies including vascular occlusion and tumour formation (Herberts et al., 2011). Furthermore, the ease with which EVs can be modified allows researchers to alter their cargo and take advantage of their role in intercellular communication to deliver specific bioactive factors, such as miRNAs, to host cells to promote neurological repair mechanisms (Alkaff et al., 2020). Autologous transplantation is also possible as EVs could be isolated from an individual, modified and returned to the body thus minimising the risk of immune rejection. Currently there have been no identified risks associated with EV administration in preclinical models, with the first clinical trial underway to assess the safety and efficacy of using allogenic MSC-derived exosomes in ischaemic stroke patients (https://clinicaltrials.gov/ct2/show/NCT03384433). It seems likely that with increasing knowledge and understanding of the biomarker and therapeutic potential of EVs and their associated miRNA cargo in AIS that their use in both these areas of current unmet clinical need could have significant impact.

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

## Additional information

### Competing interests

None.

### Author contributions

L.M.W. was responsible for the concept of the article. All authors wrote the first draft of the article, revisions following peer review and contributed to the analysis and interpretation of the data.

All authors have read and approved the final version of this manuscript and agree to be accountable for all aspects of the work in ensuring that questions related to the accuracy or integrity of any part of the work are appropriately investigated and resolved. All persons designated as authors qualify for authorship, and all those who qualify for authorship are listed.

## Funding

This work was supported by the Chief Scientist Office Project grant TCS/18/13 (L.M.W.) and the British Heart Foundation (BHF) 4 Year PhD Studentship Awards FS/18/58/34179 (R.A.R.) and FS/4yPhD/F/21/34158 (C.C.C.).

## Keywords

cargo, clinical, extracellular vesicles, ischaemic stroke, microRNAs, preclinical

## Supporting information

Additional supporting information can be found online in the Supporting Information section at the end of the HTML view of the article. Supporting information files available:

**Peer Review History**

