## [Peer Review History · The Journal of Physiology]

Extracellular vesicles and their microRNA cargo in ischaemic stroke

Josie L. Fullerton, Caitlin C. Cosgrove, Rebecca A. Rooney, and Lorraine M. Work
DOI: 10.1113/JP282050

Corresponding author(s): Lorraine Work (Lorraine.Work@glasgow.ac.uk)

Review Timeline:

Submission Date:	21-Dec-2021
Editorial Decision:	11-Feb-2022
Revision Received:	11-Mar-2022
Accepted:	15-Mar-2022

Senior Editor: Ian Forsythe

Reviewing Editor: Harold Schultz

Transaction Report:

Dear Dr Work,

Re: JP-TR-2021-282050 "Extracellular vesicles and their microRNA cargo in ischaemic stroke" by Josie L. Fullerton, Caitlin C. Cosgrove, Rebecca A. Rooney, and Lorraine M. Work

Thank you for submitting your Topical Review to The Journal of Physiology. It has been assessed by a Reviewing Editor and by 1 expert referee and I am pleased to tell you that it is considered to be acceptable for publication following satisfactory revision.

The reports are copied at the end of this email. Please address all of the points and incorporate all requested revisions, or explain in your Response to Referees why a change has not been made.

NEW POLICY: In order to improve the transparency of its peer review process The Journal of Physiology publishes online as supporting information the peer review history of all articles accepted for publication. Readers will have access to decision letters, including all Editors' comments and referee reports, for each version of the manuscript and any author responses to peer review comments. Referees can decide whether or not they wish to be named on the peer review history document.

I hope you will find the comments helpful and have no difficulty in revising your manuscript within 4 weeks.

Your revised manuscript should be submitted online using the links in Author Tasks Link Not Available. This link is to the Corresponding Author's own account, if this will cause any problems when submitting the revised version please contact us.

You should upload:

- A Word file of the complete text (including any Tables);
- An Abstract Figure, (with accompanying Legend in the article file)
- Each figure as a separate, high quality, file;
- A full Response to Referees;
- A copy of the manuscript with the changes highlighted.
- Author profile. A short biography (no more than 100 words for one author or 150 words in total for two authors) and a portrait photograph of the two leading authors on the paper. These should be uploaded, clearly labelled, with the manuscript submission. Any standard image format for the photograph is acceptable, but the resolution should be at least 300 dpi and preferably more.

- A 'Cover Art' file for consideration as the Issue's cover image;
- Appropriate Supporting Information (Video, audio or data set https://jp.msubmit.net/cgi-bin/main.plex?form_type=display_requirements#supp).

To create your 'Response to Referees' copy all the reports, including any comments from the Senior and Reviewing Editors into a Word, or similar, file and respond to each point in colour or CAPITALS. Upload this when you submit your revision.

I look forward to receiving your revised submission.

Yours sincerely,

Ian D. Forsythe
Deputy Editor-in-Chief
The Journal of Physiology
<https://jp.msubmit.net>
<http://jp.physoc.org>
The Physiological Society
Hodgkin Huxley House
30 Farringdon Lane
London, EC1R 3AW
UK
<http://www.physoc.org>
<http://journals.physoc.org>

EDITOR COMMENTS

Reviewing Editor:

This is an interesting topical review assessed by an external referee and reviewing editor. Overall the review is informative and addresses current knowledge and controversies in a rapidly changing field. As enumerated by the referee, there are areas where controversies or inconsistencies in studies are not well summarized or emphasized. The article body (Introduction to Conclusions) is a bit long (over 6000 words) with 73 references, although not overly excessive for the amount of information covered. Without sacrificing content, it should be possible to revise to reduce unnecessary wording in the narrative to keep the article below these limits. Figure 3 will require modification to correct the inconsistency described by the referee.

REFeree COMMENTS

Referee #1:

In this invited review, Josie Fullerton et al. summarize developments in the rapidly expanding field which explores diagnostic and functional significance of extracellular vesicles and microRNA in acute ischemic stroke. This contribution is timely and represents a welcome introduction for both clinical and translational scientists who are interested in relevance of EVs and miRs. A particular value of this exercise is that the Authors draw a parallel between clinical and model/translational findings. As I read the review, I accumulated a number of critical questions, many of which have been addressed by the section "Commonalities in EV-packaged miRNA expression changes from bench to bedside".

The manuscript comes with caveats that mirror limitations of the EV/miR field as a whole: (i) the Authors recite many findings without their critical deconstruction and cross-referencing, (ii) some of the statements do not fully represent experimental observations, (iii) a number of conflicting findings are not reconciled. My suggestions for improvements in the manuscript are enumerated below:

Specific comments:

[1] In the Abstract, the following statement is simplistic and potentially misleading: "The majority of circulating miRNAs are actively encapsulated by extracellular vesicles (EVs) and mesenchymal stem cells (MSCs) are the most prolific producers of EVs." It is true that "therapeutics" EVs are produced in bulk in the cultures of MSC. However, the endogenous EVs (including those used for diagnostic purposes) are produced by many other cell types.

[2] For the General Reader, please explain the -3p/5p specification of miR species in the "MicroRNAs" subsection.

[3] In the Introduction, it would be helpful to mention that besides miRs, EVs can deliver additional beneficial substances, such as membrane-bound proteins, cytosolic metabolites and anti-apoptotic proteins, as well as mitochondria. This well-established concept is touched upon in Figure 2, but not addressed in the text.

[4] Even in this review, I can identify several instances, when mesenchymal stem cell derived EVs lose their protective properties upon knockdown of individual miRs (e.g., miR-124 in Song et al. 2019; miR-138-5p in Deng et al., 2019; miRNA-146a in Wang et al., 2021). These studies contradict each other, because they suggest that all "other" miR species are irrelevant or play a minor role. I understand the Authors' uneasy task to provide a concise summary of the field. Still, some relevant discussion would be appropriate.

[5] The literature findings on potential cellular targets for EV/miR actions are somewhat confusing: target cell types range from vasculature to glial cells to neurons. For the General Reader, it may be beneficial to additionally summarize (either in Introduction or Conclusions) what cell types can be targeted by EVs and provide expert opinion on the relevant mechanisms of delivery.

[6] There are some inconsistencies between literature findings and their description in the text. As one example, miR-124-3p is characterized as upregulated in ischemic stroke. On surface, this statement is supported by the Authors' findings (table in

J.C. van Kralingen et al., 2019) and the results of Q. Ji et al. (PLOS ONE, 2016). Yet, Z. Qi et al. found the opposite trend in clinical samples a decrease in MIR-124-3p levels in stroke patients (Front Mol Biosci, 2021). The preclinical therapeutic study of Li et al. (Theranostics, 2021) also demonstrated a drop in miR-124-3p, but now in a mouse brain tissue. These discrepancies are not fully acknowledged and discussed in the text. Please flesh them out, whenever possible.

[7] Figure 3 shows several miRs as both upregulated and downregulated in clinical stroke (miR-30a, miR-124, and miR-422a). This conflicting information is explained in the text but needs to be additionally addressed in the figure and figure legend via references to AIS, HIS, etc.

[8] Personally, I always had a hard time with understanding why "the systemic administration of EVs, rather than stem cells for example, would be advantageous as their small size allows them to readily cross the BBB". Lipophilic EVs are supposed to be trapped by the endothelial cells. Is there a specific mechanism for cross-endothelial delivery of EVs, besides partial opening of BBB in post-ischemic tissue? Perhaps the Authors can elaborate more on this topic.

REQUIRED ITEMS:

-Author profile(s) must be uploaded via the submission form. Authors should submit a short biography (no more than 100 words for one author or 150 words in total for two authors) and a portrait photograph of the two leading authors on the paper. These should be uploaded, clearly labelled, with the manuscript submission. Any standard image format for the photograph is acceptable, but the resolution should be at least 300 dpi and preferably more. A group photograph of all authors is also acceptable, providing the biography for the whole group does not exceed 150 words.

END OF COMMENTS

Confidential Review

21-Dec-2021

11th March 2022

Professor Ian Forsythe
Deputy Editor in Chief, J Physiol

Dear Professor Forsythe and the Editorial Board of the Journal of Physiology

Ms. Ref. No.: JP-TR-2021-282050

I refer to your email dated 11th February 2022 in relation to our recently submitted review article "Extracellular vesicles and their microRNA cargo in ischaemic stroke" by Fullerton *et al* and your decision that you are willing to consider a revised manuscript following revision in light of comments from reviewers. We would like to thank the reviewers for their positive and supportive comments about the merit and value in the review. We have addressed all the comments made by the reviewers, providing individual responses below. Relevant changes to the text in the manuscript have been highlighted in red. We believe the input from the reviewer's has improved the review article and we hope it is now of a quality deemed suitable for publication in the *Journal of Physiology*.

EDITOR COMMENTS

Reviewing Editor:

This is an interesting topical review assessed by an external referee and reviewing editor. Overall the review is informative and addresses current knowledge and controversies in a rapidly changing field. As enumerated by the referee, there are areas where controversies or inconsistencies in studies are not well summarized or emphasized. The article body (Introduction to Conclusions) is a bit long (over 6000 words) with 73 references, although not overly excessive for the amount of information covered. Without sacrificing content, it should be possible to revise to reduce unnecessary wording in the narrative to keep the article below these limits. Figure 3 will require modification to correct the inconsistency described by the referee.

We have amended the text of the review broadly to try and reduce the wordcount. However, as we were also required to add in further detail in places to address the comments of reviewer #1, this has resulted in the overall wordcount now being around 6200 (introduction – conclusions). Changes to the legend for Figure 3 have been made to address the comments of reviewer #1 and the editor, these are detailed below.

REFEREE COMMENTS

Referee #1:

In this invited review, Josie Fullerton et al. summarize developments in the rapidly expanding field which explores diagnostic and functional significance of extracellular vesicles and microRNA in acute ischemic stroke. This contribution is timely and represents a welcome introduction for both clinical and translational scientists who are interested in relevance of EVs and miRs. A particular value of this exercise is that the Authors draw a parallel between clinical and model/translational findings. As I read the review, I accumulated a number of critical questions, many of which have been addressed by the section "Commonalities in EV-packaged miRNA expression changes from bench to bedside".

The manuscript comes with caveats that mirror limitations of the EV/miR field as a whole: (i) the Authors recite many findings without their critical deconstruction and cross-referencing, (ii) some of the statements do not fully represent experimental observations, (iii) a number of conflicting findings are not reconciled. My suggestions for improvements in the manuscript are enumerated below:

Specific comments:

[1] In the Abstract, the following statement is simplistic and potentially misleading: "The majority of circulating miRNAs are actively encapsulated by extracellular vesicles (EVs) and mesenchymal stem cells (MSCs) are the most prolific producers of EVs." It is true that "therapeutics" EVs are produced in bulk in the cultures of MSC. However, the endogenous EVs (including those used for diagnostic purposes) are produced by many other cell types.

We have amended the abstract to address this comment as follows *"...(EVs) produced by many cells and organs endogenously. EVs released by mesenchymal stem cells (MSCs) have been extensively studied for their therapeutic potential."* We trust that this will be to the reviewer's satisfaction.

[2] For the General Reader, please explain the -3p/5p specification of miR species in the "MicroRNAs" subsection.

We have amended the text in this section to include the sentence *"Two mature miRNA species can be generated from the 5' and 3' arms of pre-miRNA, termed miRNA-5p/3p. Predominantly, one species remains while the other is degraded, yet co-existence of both species is increasingly reported."* We trust that this will be to the reviewer's satisfaction.

[3] In the Introduction, it would be helpful to mention that besides miRs, EVs can deliver additional beneficial substances, such membrane-bound proteins, cytosolic metabolites and anti-apoptotic proteins, as well as mitochondria. This well-established concept is touched upon in Figure 2, but not addressed in the text.

We have amended the text in the introduction to include these substances as recommended by the reviewer. This now reads *"...membrane-bound proteins, lipids, receptors, cytosolic metabolites, cytosolic and anti-apoptotic proteins, such as mitochondria."*

[4] Even in this review, I can identify several instances, when mesenchymal stem cell derived EVs lose their protective properties upon knockdown of individual miRs (e.g., miR-124 in Song et al. 2019; miR-138-5p in Deng et al., 2019; miRNA-146a in Wang et al., 2021). These studies contradict each other, because they suggest that all "other" miR species are irrelevant or play a minor role. I understand the Authors' uneasy task to provide a concise summary of the field. Still, some relevant discussion would be appropriate.

We have added a short sentence to the end of the MSC EV subsection to highlight this important consideration raised by the reviewer. However, we have cited alternative studies to those highlighted here as the Song and Wang studies were not using MSC-derived EVs. We trust that this will be to the satisfaction of the reviewer. *"Additionally, the combinatory role of several miRNAs to promote recovery should not be undermined despite findings that MSC-EVs lose their protective abilities upon the knockdown of individual miRNAs (Xin et al. 2013, Deng et al. 2019, Zhang et al. 2021)"*.

[5] The literature findings on potential cellular targets for EV/miR actions are somewhat confusing: target cell types range from vasculature to glial cells to neurons. For the General Reader, it may be beneficial to additionally summarize (either in Introduction or Conclusions) what cell types can be targeted by EVs and provide expert opinion on the relevant mechanisms of delivery.

We have added the following paragraph and supporting citations to the subsection on EVs. We hope that this addresses this recommendation from the reviewer as this was rather challenging to encompass succinctly. *"Ever-expanding studies indicate that EVs target recipient cells such as, natural killer cells, B- and T- cells, fibroblasts, endothelial cells, monocytes, macrophages, lymphocytes, adipocytes, hepatocytes, dendritic cells, cancer cells and glioblastomas, neural cells, microglia, astrocytes, and oligodendrocytes. The majority of evidence suggest that EVs are internalised into these recipient cells by endocytosis; however, the exact mechanism of EV uptake remains unclear. Suggested mechanisms include clathrin-mediated endocytosis, caveolin-dependent endocytosis, lipid-raft mediated endocytosis, macro- or micropinocytosis and phagocytosis; this uptake leads to subsequent release of EV contents from originating cells into recipient cells"*.

[6] There are some inconsistencies between literature findings and their description in the text. As one example, miR-124-3p is characterized as upregulated in ischemic stroke. On surface, this statement is supported by the Authors' findings (table in J.C. van Kralingen et al., 2019) and the results of Q. Ji et al. (PLOS ONE, 2016). Yet, Z. Qi et al. found the opposite trend in clinical samples a decrease in MIR-124-3p levels in stroke patients (Front Mol Biosci, 2021). The preclinical therapeutic study of Li et al. (Theranostics, 2021) also demonstrated a drop in miR-124-3p, but now in a mouse brain tissue. These discrepancies are not fully acknowledged and discussed in the text. Please flesh them out, whenever possible.

We have amended the text in the clinical sections to address this request through the addition of *"There are multiple factors that could influence the differential expression of miRNA-124 between studies, primarily the specific targeting of the 3' strand of miRNA-124 by Qi et al. (2021), but also sample collection post-stroke (2hrs vs 24hrs), patient characteristics, or patient comorbidities."* We respectfully draw the reviewer's attention to the Song et al (2019) and Li et al (2021) studies which each support the findings of the other: that knockdown of miRNA-124 in M2-derived EVs leads to a poorer outcome and that the proposed mechanism of M2-derived EVs is mediated, at least in part, by miRNA-124. We trust that this change and clarification will be to the satisfaction of the reviewer.

[7] Figure 3 shows several miRs as both upregulated and downregulated in clinical stroke (miR-30a, miR-124, and miR-422a). This conflicting information is explained in the text but needs to be additionally addressed in the figure and figure legend via references to AIS, HIS, etc.

We have amended both the figure legend in light of the reviewers comments to ensure this matches which is depicted in the figure and described in the text.

[8] Personally, I always had a hard time with understanding why "the systemic administration of EVs, rather than stem cells for example, would be advantageous as their small size allows them to readily cross the BBB". Lipophilic EVs are supposed to be trapped by the endothelial cells. Is there a specific mechanism for cross-endothelial delivery of EVs, besides partial opening of BBB in post-ischemic tissue? Perhaps the Authors can elaborate more on this topic.

We have added some text to try and address this important consideration to the conclusion *"The systemic administration of EVs, rather than stem cells for example, has the major advantage of being able to deliver the therapeutic agent across the BBB (Heidarzadeh et al., 2021). The exact mechanism of this ability is largely unknown but recent studies have highlighted a diverse range of uptake strategies across the intact BBB including the use of specific transporters, initiators of adsorptive transcytosis, receptor interactions, and a notable influence from the brain-to-blood efflux system. Additionally, inflammation, such as that in stroke, breaks down the BBB and therefore increases its permeability which may further enhance these transport processes (Banks et al., 2020)."* Further, we removed the reference to the BBB permeability in the earlier MSC section.

We hope that by addressing the points raised by the referees, we have improved the quality of the review overall by adding critique and editing the text while not sacrificing content. We look forward to hearing from you in this regard.

Yours sincerely

(On behalf of all authors)

Dr Lorraine M. Work, PhD BSc (Hons)

Reader

Institute of Cardiovascular & Medical Sciences

BHF Glasgow Cardiovascular Research Centre

University of Glasgow

126 University Place

Glasgow G12 8TA

Tel: 0141 330 5869

Email: Lorraine.Work@glasgow.ac.uk

Dear Dr Work,

Re: JP-TR-2022-282050R1 "Extracellular vesicles and their microRNA cargo in ischaemic stroke" by Josie L. Fullerton, Caitlin C. Cosgrove, Rebecca A. Rooney, and Lorraine M. Work

I am pleased to tell you that your Topical Review article has been accepted for publication in The Journal of Physiology, subject to any modifications to the text that may be required by the Journal Office to conform to House rules.

IMPORTANT

We seem to be missing legend to accompany your abstract figure. Please can you send this as a Word file to the JP office as soon as possible: jp@physoc.org

Thank you!

NEW POLICY: In order to improve the transparency of its peer review process The Journal of Physiology publishes online as supporting information the peer review history of all articles accepted for publication. Readers will have access to decision letters, including all Editors' comments and referee reports, for each version of the manuscript and any author responses to peer review comments. Referees can decide whether or not they wish to be named on the peer review history document.

The last Word version of the paper submitted will be used by the Production Editors to prepare your proof. When this is ready you will receive an email containing a link to Wiley's Online Proofing System. The proof should be checked and corrected as quickly as possible.

All queries at proof stage should be sent to tjp@wiley.com

The accepted version of the manuscript will be published online, prior to copy editing in the Accepted Articles section.

Are you on Twitter? Once your paper is online, why not share your achievement with your followers. Please tag The Journal (@jphysiol) in any tweets and we will share your accepted paper with our 22,000+ followers!

Yours sincerely,

Ian D. Forsythe
Deputy Editor-in-Chief
The Journal of Physiology
<https://jp.msubmit.net>
<http://jp.physoc.org>
The Physiological Society
Hodgkin Huxley House
30 Farringdon Lane
London, EC1R 3AW
UK
<http://www.physoc.org>
<http://journals.physoc.org>

* IMPORTANT NOTICE ABOUT OPEN ACCESS *

Information about Open Access policies can be found here <https://physoc.onlinelibrary.wiley.com/hub/access-policies>

To assist authors whose funding agencies mandate public access to published research findings sooner than 12 months after publication The Journal of Physiology allows authors to pay an open access (OA) fee to have their papers made freely available immediately on publication.

You will receive an email from Wiley with details on how to register or log-in to Wiley Authors Services where you will be able to place an OnlineOpen order.

You can check if your funder or institution has a Wiley Open Access Account here <https://authorservices.wiley.com/author-resources/Journal-Authors/licensing-and-open-access/open-access/author-compliance-tool.html>

Your article will be made Open Access upon publication, or as soon as payment is received.

If you wish to put your paper on an OA website such as PMC or UKPMC or your institutional repository within 12 months of publication you must pay the open access fee, which covers the cost of publication.

OnlineOpen articles are deposited in PubMed Central (PMC) and PMC mirror sites. Authors of OnlineOpen articles are permitted to post the final, published PDF of their article on a website, institutional repository, or other free public server, immediately on publication.

Note to NIH-funded authors: The Journal of Physiology is published on PMC 12 months after publication, NIH-funded authors DO NOT NEED to pay to publish and DO NOT NEED to post their accepted papers on PMC.

EDITOR COMMENTS

Reviewing Editor:

Thank you for a well written and informative review.

REFEREE COMMENTS

Referee #1:

In this invited review, Josie Fullerton et al. summarize developments in the rapidly expanding field which explores diagnostic and functional significance of extracellular vesicles and microRNA in acute ischemic stroke. This contribution is timely and represents a welcome introduction for both clinical and translational scientists who are interested in relevance of EVs and miRs.

The revised version of the manuscript addressed previously noted concerns.

1st Confidential Review

11-Mar-2022